# Halogens in Acetophenones Direct the Hydrogen Bond Docking Preference of Phenol via Stacking Interactions

**DOI:** 10.3390/molecules26164883

**Published:** 2021-08-12

**Authors:** Charlotte Zimmermann, Manuel Lange, Martin A. Suhm

**Affiliations:** Institut für Physikalische Chemie, Georg-August-Universität Göttingen, Tammannstr. 6, 37077 Göttingen, Germany; czimmer2@gwdg.de (C.Z.); mlange6@gwdg.de (M.L.)

**Keywords:** London dispersion, ketone complexes, density functional theory, hydrogen bonds, molecular recognition, vibrational spectroscopy, gas phase, benchmark

## Abstract

Phenol is added to acetophenone (methyl phenyl ketone) and to six of its halogenated derivatives in a supersonic jet expansion to determine the hydrogen bonding preference of the cold and isolated 1:1 complexes by linear infrared spectroscopy. Halogenation is found to have a pronounced effect on the docking site in this intermolecular ketone balance experiment. The spectra unambiguously decide between competing variants of phenyl group stacking due to their differences in hydrogen bond strength. Structures where the phenyl group interaction strongly distorts the hydrogen bond are more difficult to quantify in the experiment. For unsubstituted acetophenone, phenol clearly prefers the methyl side despite a predicted sub-kJ/mol advantage that is nearly independent of zero-point vibrational energy, turning this complex into a challenging benchmark system for electronic structure methods, which include long range dispersion interactions in some way.

## 1. Introduction

The stacking energetics of aromatic rings is of paramount importance for the materials and life sciences [1,2,3,4,5]. Any electronic structure method that is described as relevant in these fields must be capable of reproducing such stacking interaction strengths, also in competition with other intermolecular interactions. This is quite independent on the actual explanation for the dominance of such aromatic pairings [6,7,8]. Ultimately, the stacking energetics should be checked experimentally for model systems in the gas phase at low temperature to remove any complexity of the solvent environment and thermal excitation, but spectroscopic tools to determine gas phase interaction energies are scarce [9]. Spectroscopy is much better in determining interaction-induced frequency shifts and structures, which are only indirectly related to the energetics.

To help remove this bottleneck, we have proposed the study of intermolecular energy balances by vibrational spectroscopy in the gas phase [10]. A hydrogen bond donor docks onto a divalent hydrogen bond acceptor and its docking preference between the two binding sites reflects not only the local bonding situation but also the interaction between more distant parts of the two molecules, which may come close to each other. Ketones as acceptors offer an important advantage [10,11,12]. They have two locally almost equivalent docking sites, the two lone electron pairs at the carbonyl oxygen, between which the donor molecule can switch easily. This ensures that any interaction difference between the two lone pairs is largely determined by secondary interactions and any zero-point vibrational energy (ZPVE) correction between the sites is minimized, because it involves a similar local hydrogen bond environment. The latter assumption may break down if the secondary interaction becomes so strong that it seriously affects the hydrogen bond strength, in other words, if it becomes the primary interaction and competes with hydrogen bonding. In this limit, zero-point energy issues again become important, but the expectation is that they are still captured reasonably well in the harmonic approximation, as long as some hydrogen bonding persists.

In the present work, we combine the simplest aromatic ketone, acetophenone, with the simplest aromatic hydroxy compound, phenol, to weigh the interaction between the two aromatic rings against the alternative structure, where the phenyl ring of phenol weakly interacts with the methyl group on the ketone. This is illustrated in Figure 1 with a schematic view along the CH${}_{3}$-C=O plane toward the more or less tilted aromatic part of the ketone. Hydrogens or halogens bound to carbon atoms are suppressed for simplicity and generality. In the methyl docking variant (Me), the phenol pointing to the observer realises an almost in-plane hydrogen bond. In the phenyl docking variant (Ph), the phenol points away from the observer and realizes a partial alignment with the aromatic part of the acetophenone, but the hydrogen bond is clearly dominant. In the Ph${}_{S}$ structure, stacking forces dominate, partly assisted by a twist in the acetophenone, and the hydrogen bond is distorted away from the ketone plane. These three docking options are explored as a function of acetophenone halogenation. By using an intermolecular hydrogen bond link, we ensure that the attractive or repulsive interactions are only mildly distorted by the connecting hinge. This requires low temperatures, realized in supersonic jet expansions, which at the same time minimize entropic contributions and allow to focus on the energetics of the stacking interaction. To avoid any perturbation of the conformational equilibrium by the environment, the complexes are prepared in vacuum, rather than in embedded form, such as in cryogenic matrix isolation techniques [13]. In contrast to earlier solution studies [14], which had to rely largely on empirical correlations, the present results allow for direct comparison with quantum chemical predictions.

Acetophenone is well suited for vibrational spectroscopy because the mixed sp${}^{2}$/sp${}^{3}$ substitution at the keto group allows for a different hydrogen bond angle due to donor oxygen interaction with the α-CH groups in the asymmetrically substituted ketone [11]. Therefore, phenyl and methyl side docking induce distinct, easily assignable OH stretching vibrations in the two isomers. The spectral intensity, in particular when combined with robust theoretical predictions of infrared absorption cross section, reflects the relative abundance of the two docking isomers [15]. Due to the nonequilibrium nature of the supersonic jet expansion, the equilibration between the two isomers freezes below some effective conformational temperature [10]. Depending on the detailed system, this can happen anywhere between the nozzle temperature (typically 300 K or higher) and the rotational temperature of the complex after expansion into the vacuum (down to 10 K). The low but finite interconversion barrier for ketones makes values between roughly 40 and 160 K more likely, but strong aromatic interaction could lead to earlier freezing. In the case of water as donor, depending on the preparation and detection technique, one [16] or two [17] isomers can be found.

In the present work, we show that the prototype balance involving phenol and plain acetophenone works as expected. We show that halogen-substituted acetophenones in the 2- and 4-positions largely confirm the predicted trends at dispersion-corrected DFT level, with a few exceptions for fluorine substitution, which we partially blame on the subtle failure of our simplest computational model. This is supported by small modifications of the model and its predictions. One challenge is that the spectral visibility of isomers with severely weakened hydrogen bond decreases compared to those with a strong primary hydrogen bond interaction. This weakens the quantitative conclusions about the relative abundance of docking isomers despite clear-cut spectral patterns. Halogen substitution in the 3-position was not explored because it leads to subtle conformational isomerism and would complicate the picture.

## 2. Materials and Methods

### 2.1. Experimental Methods

The investigated halogenated acetophenone derivatives are abbreviated *n*X, where X stands for the halogen (F, Cl, Br) and *n* denotes the aromatic ring substitution in position 2 (*ortho*), 4 (*para*), or 0 (unsubstituted). Without further purification, gaseous 0F (Sigma-Aldrich, Taufkirchen, Germany, 99%), 2F (ACROS organics, Schwerte, Germany, 97%), 4F (ACROS organics, 99%), 2Cl (Sigma-Aldrich, 97%), 4Cl (Sigma-Aldrich, 97%), and 2Br (Sigma-Aldrich, 99%) were each mixed with phenol (Alfa Aesar, Kandel, Germany, ≥99%) in a large excess of helium (Linde, Pullach, Germany, 99.996%) and admitted from a 67 L reservoir at 0.75 bar or 1.25 bar through six magnetic valves into a pre-expansion chamber. From there, the gas expanded through a 600 mm×0.2 mm slit nozzle into a vacuum chamber. The background pressure was kept low by large buffer volumes and by continuous pumping at 500 to 2500 $m^{3}$/h while the expansion was probed using a Bruker 66v/S FTIR spectrometer with a resolution of 2 ${\mathrm{cm}}^{-1}$, ${\mathrm{CaF}}_{2}$ optics and a 150 W tungsten lamp. The IR absorption was detected using a liquid nitrogen cooled InSb detector and coaveraged over 225 to 400 pulses to obtain spectra in the OH/CH stretching region. To differentiate between alcohol monomers and mixed complexes, the concentrations were varied. Sufficient dilution of the phenol minimized the abundance of homodimers and oligomers. Further details on the experiment can be found elsewhere [18].

Due to its low volatility, complexes of 4Br (Sigma-Aldrich, 98%) with phenol (Alfa Aesar, ≥99%) were probed using a similar spectrometer and a heatable setup with a 60 mm×0.2 mm slit nozzle. Details of this setup (“popcorn-jet”) can be found elsewhere [19,20]. Since both substances needed to be heated to different temperatures, two heatable sample chambers were placed in the gas flow (“double pick-up”) for the first time in the present work. More information about this new extension is available in the Supporting Information (Appendix A). 25 double-sided scans with appropriate concentration ratio of phenol, 4Br, and He were coaveraged, as discussed in the Supporting Information.

Neither of the techniques is strictly size selective, but concentration variation usually allows to identify and discriminate 1:1 isomers from other cluster compositions if the absorption signal is sufficiently high. For this work, signal intensity ratios between isomers are particularly important because they allow for conclusions about methyl-to-phenyl docking isomer ratios. These experimental band integral ratios $\frac{I_{\mathrm{Me}}}{I_{\mathrm{Ph}}}$ were determined by a modified automated statistical evaluation [21] where the positions of the band maxima and a statistical variation for the integration range (here chosen between 6 to 20 ${\mathrm{cm}}^{-1}$, depending on the spectrum) are included as parameters [22]. Synthetic noise with characteristics comparable to the experimental one is added to obtain a reliable statistical error bar for the relevant intensities and intensity ratios $\frac{I_{\mathrm{Me}}}{I_{\mathrm{Ph}}}$.

### 2.2. Computational Methods

Due to the experimental focus of this study, results from DFT methods were mainly used for band assignment and isomer quantification purposes. Further theory benchmarking in terms of their ability to robustly describe the combination of hydrogen bonding, London dispersion and the influence of substitution is left for the future. Therefore, this study is limited to structure optimizations on B3LYP-D3/def2-TZVP level [23,24,25,26] including D3 dispersion correction [27] with Becke–Johnson damping [28,29,30,31] and a three-body term [32]. For the initial manual structure search, which was additionally backed up using the Crest tool [33], ORCA version 4.2.1 [34] was used. Reoptimization was carried out with the minimally augmented ma-def2-TZVP basis set [35]. Reaction path optimizations and some relaxed scans were carried out using Turbomole [36,37]. Computational details are listed in Appendix A in the Supporting Information. Given the low but mode-dependent temperatures during a jet expansion (approximately 10 K for rotation, approximately 100 K for vibration), thermal corrections were neglected. In favorable cases and building on theoretical predictions of the energy difference between isomers, the isomer ratio can be directly interpreted in terms of a Boltzmann distribution with a conformational temperature $T_{c}$ at which the jet cooling is frozen in due to the interconversion barrier [10]. Harmonic zero-point vibrational energy (ZPVE) was included in the analysis, although it largely cancels between the isomers of a carbonyl balance due to the two lone electron pairs being similar in their anisotropy toward a hydrogen bond donor [11]. The cancelation is less perfect if one of the sides involves a strongly distorted hydrogen bond due to distant interactions.

## 3. Results and Discussion

Figure 2 shows the infrared spectrum of phenol in the OH stretching region, when expanded together with acetophenone (0F) in helium through a 600 mm slit nozzle. The homodimer (PhOH)${}_{2}$ [38] is downshifted relative to the monomer (PhOH, 3657 ${\mathrm{cm}}^{-1}$ [39]) due to the hydrogen bond formation. Further downshifted are two signals, which can be assigned to the phenyl- (O${}_{\mathrm{Ph}}$) and methyl-side (O${}_{\mathrm{Me}}$) docking of phenol to acetophenone. Judging by the difference in downshift, the ketone is a more attractive hydrogen bond acceptor for phenol than phenol itself. The coordination angle at the carbonyl group is more favorable on the less bulky methyl side, thus leading to a larger shift.

This spectral separation of phenyl and methyl docking on the order of 20% of the total shift persists for the halogenated acetophenones (symbols above the peaks for the parent complex, see Appendix A in the Supporting Information for the corresponding spectra) and thus leads to a straightforward experimental docking assignment. The downshifts are quite sizeable, but in contrast to the hydrate case [17], there is no pronounced resonance coupling pattern evident in the spectra, thus facilitating the assignment. The monomer C=O overtone is predicted/found close to 3400 ${\mathrm{cm}}^{-1}$ (see Appendix A and [40] for 2F). In some cases, it overlaps with the methyl docking signal of the complex; in others, it does not. This situation and its negligible consequences for the intensity analysis are discussed in the Supporting Information for each case (Appendix A).

The much weaker signal for (O${}_{\mathrm{Ph}}$) suggests a preference for methyl docking, but theoretical cross sections will be needed for a more quantitative assessment. While fluorination conserves the simultaneous detection of both docking isomers, chlorine and bromine substitution suggests a clear preference for either isomer, depending on the substitution position. This already hints at the sensitivity of the present approach to probe docking preferences.

Figure 3 correlates the experimental band positions and their splitting due to docking variants with harmonic predictions for the most stable docking structures predicted at the standard B3LYP-D3/def2-TZVP level. As expected, the harmonic shift predictions overestimate the experimental (anharmonic) values. This is more a DFT deficiency than a consequence of the harmonic approximation [41,42]. The diagonal line indicates negligible overestimation, which is the case when the splittings between Me and Ph docking are looked at (half-filled symbols), due to error compensation between two similarly overestimated shifts. The absolute shifts of the strong Me docking signals show a uniform scaling and most of the Ph docking wavenumbers cluster together. There are two prominent outliers, namely 2F and 4F with Ph docking. Here, the experimental shifts for the weaker IR signals match the harmonic prediction quite closely, but this is clearly for the wrong reason. A formal remedy that was successful for other substituted acetophenones [11] is to look for a secondary, energetically close isomer which better fits the theory–experiment comparison. Sometimes, a combination of DFT deficiency and basis set size incompleteness leads to a close competition of two such isomers with a small interconversion barrier in between. Our experiment is then quite sensitive to the relative energy of the two (not necessarily both real) structures and points at the more stable structure via the observed hydrogen bond shift. Here, the interpretation would be that calculated fluorinated acetophenone structures on the Ph side lead to too much stacking interaction with phenol at the B3LYP-D3/def2-TZVP level, at the expense of the hydrogen bond between the phenol and the keto group. The predicted secondary structures (2F’, 4F’) with less stacking and more hydrogen bonding indeed fit experiment better and are more likely candidates for the most stable Ph docking structures of the fluorinated acetophenone complexes.

To explore whether this theory deficiency is dominated by basis set incompleteness or by DFT limitations, we have repeated the calculations with a minimally augmented basis set. The structural changes are minor (see Supporting Information), but indeed 2F and 2F’ now switch their relative energy sequence after ZPVE correction (see Appendix A) and the new global minimum structure 2F’ fits the correlation of the other halogenated acetophenones well (Figure 4). Note that the effects are subtle and in this case depend on the inclusion of ZPVE despite the canceling strategy described in the introduction. Therefore, 2F/2F’ represents a sensitive test case for theory based on the spectroscopic sensitivity to the stacking geometry. For 4F, the situation remains contradictory between the predicted energy sequence and the experimentally observed band, but again, the prediction depends on ZPVE correction for the smaller basis set. Clearly, the observed phenyl docking signal is not due to the 4F structure with pronounced aromatic stacking (Figure 4). Instead, it is more likely to be due to 4F’, which is predicted about 0.5 $\mathrm{kJ}/{\mathrm{mol}}^{-1}$ less stable after harmonic ZPVE correction for both employed basis sets. Considering that the two structures differ significantly in hydrogen bond strength, this is within the uncertainty of the nuclear quantum correction. At this stage, one should not completely rule out the coexistence of the 2F and 4F phenyl docking structures in the expansion because there is evidence for some weaker bands (vide infra), but they are clearly not responsible for the observed strong bands.

After having firmly established the assignment of the main IR features to different docking isomers, including the need for at least minimally augmented TZ basis sets to reproduce most of the experimental trends, we can turn to the quantification of the docking preference. An essential theory input at this stage is the ratio of the IR absorption cross sections or band strengths. As in the case of fundamental frequencies, it is not crucial to obtain the correct absolute values, which would require anharmonic treatment. Instead, it may be safely assumed that the harmonic prediction for the ratio between the docking isomers is robust, as anharmonicity effects between the similar docking sites are likely to cancel to a sufficient extent, as in the case of spectral shifts. This assumption might break down in case of a strongly competing secondary interaction, which weakens the hydrogen bond. Interestingly, the cross sections for the two docking sides differ widely (Appendix A). Whereas the Me docking isomer has a largely substitution-independent band strength in the range of 1000 to 1300 km ${\mathrm{mol}}^{-1}$, the Ph docking band strengths are much weaker and cover an unusually wide range of 200 to 800 km ${\mathrm{mol}}^{-1}$. Figure 5 helps to understand this strong substitution dependence by plotting the Ph-Me difference between the out-of-plane tilting angles of the hydrogen-bonded H relative to the ketone plane for the two isomers vs. the Ph/Me IR band strength ratio. Clearly, a low value of this band strength ratio correlates with the prediction of strong out-of-plane tilting on the phenyl side and a high value correlates with largely in-plane coordination on both sides. This is consistent with expectations for a strong hydrogen bond anisotropy of the C=O group. Any competing interaction that induces a deviation from planar coordination of the ketone not only leads to a reduction of the OH downshift (Figure 3 and Figure 4) but also to a reduction of the band strength (Figure 5). The only noticeable outliers to this trend are 2Cl and 2Br, but the reason becomes clear when inspecting the intramolecular torsion of the phenyl substituent relative to the keto group (Appendix A). Due to the *ortho* repulsion of the bulky halogens with the methyl group, the phenyl group is tilted out of the keto plane already in the monomer. This allows for favorable stacking interaction with phenol, but it also affects the conjugation of the keto group with the aromatic ring, and thus expectedly the acceptor strength.

Because basis set augmentation has little effect on the theoretical band strength ratios (Appendix A), these ratios can be used to extract reasonably robust relative abundances of the two docking isomers in experiment. We start with 2Cl (2Br), where the experimental spectrum shows a single band, which is exclusively due to phenyl-side docking. A methyl docking signal of more than 25% (27%) can be strictly ruled out (see Appendix A). Because phenyl docking has the lower IR visibility (Figure 5), this translates into more than 89% (88%) phenyl docking isomer in the expansion. For 4Cl (4Br), where the experiment also shows a single signal, now due to methyl docking, the situation is less clear. While any phenyl docking signal will be a least 4.2-(2.7-) fold weaker, its lower visibility is still compatible with up to 62% (72%) phenyl docking abundance. In other words, the limited signal-to-noise ratio and low visibility do not allow to state which of the isomers is more abundant in this case. Methyl docking might be completely dominant, but it could also be slightly inferior to phenyl docking. The numbers are nearly invariant with the basis set, suggesting that the theoretical intensity input is not decisive.

The fluorinated acetophenones represent more interesting cases. Here, the methyl docking signal is clearly the strongest and the weaker phenyl docking signal is inconsistent with strong stacking (Appendix A) but instead must be due to a stacking variant that conserves more of the hydrogen bonding (e.g., 2F’, 4F’). When the smaller IR cross sections for phenyl docking are taken into account, this translates into 48 to 59% phenyl docking for 2F’ and only 19 to 30% phenyl docking for 4F’, competing with methyl docking. However, in both spectra, there are weak signals near the position where the strong stacking structures 2F and 4F are expected to absorb. These have low IR visibilities (Appendix A) and if their abundance is generously estimated from the peak integral or the noise level, they might amount to as much as about 50% in competition with methyl docking, although they are hardly visible. Table 1 summarizes the relative abundance bounds for the three different kinds of docking structures, Me, Ph (stacking competes with hydrogen bonding), and Ph${}_{S}$ (strong stacking), which result from these pairwise abundance estimates. Clear-cut statements about the global minimum structure can be made for 0F (Me) and 2Cl, 2Br (Ph${}_{S}$). For 2F, there could be a similar abundance of all three species, but Ph${}_{S}$ is the least likely to represent the global minimum. In the cases of 4F, 4Cl, and 4Br, Me and Ph${}_{S}$ docking compete for the global minimum. These experimental constraints are consistent with the theoretical B3LYP predictions for both basis sets, with 2F for the small basis set being the only (subtle) exception. The combined experimental and computational findings as a function of halogenation are summarized in Figure 6. The theoretical predictions largely correlate with the observations, with the exception of 4Cl and 4Br, where the low IR visibility and limited signal-to-noise ratio presumably hides the predicted global minimum structures.

In the cases of 0F, 2Cl, and 2Br, conformational temperatures $T_{c}$ can be estimated based on the theoretical predictions for the energy difference between two competing conformations. For 2Cl and 2Br, the computed energy advantage of phenyl docking is so large that only an upper bound for the conformational temperature higher than room temperature (less than about 400 to 600 K) can be derived. For 0F, the conformational temperature is (53±10) K (based on the smaller basis set) and (46±9) K (based on the augmented basis set). These are plausible values, perhaps at the lower end of the expected range. This lower end location indicates that the computed energy advantage for Me docking is also at the lower end of the experimentally compatible value. It will be interesting to see whether higher level calculations support a larger energy gap between the two competing structures.

## 4. Conclusions

The complexes of phenol with halogenated acetophenones show a wide range of energy preferences for either the phenyl or the methyl side of the ketone. For phenyl docking, there is a variable and sometimes close competition between stacking- and hydrogen-bond-dominated structures. This also leads to large variations in the infrared visibility of the complexes, which profits from hydrogen bonding and suffers from stacking-induced distortion of the hydrogen bond. *Ortho*-substitution prepares the acetophenone for simultaneous stacking and hydrogen bonding by tilting the ketone plane against the aromatic plane already in the monomer.

The infrared spectra find a clear-cut interpretation for the strong transitions. The band position can unambiguously distinguish between stacking- and hydrogen-bond-dominated structures on the phenyl side, even if these are close in energy. However, the low visibility of the strong stacking variants complicates the interpretation of the spectra in terms of abundance.

The phenol complex with nonhalogenated acetophenone allows for the most quantitative conclusions. Here, methyl docking strongly dominates in the spectrum despite a subtle energy advantage of only 0.5 to 0.9 $\mathrm{kJ}/{\mathrm{mol}}^{-1}$. This energy advantage is insensitive to zero-point energy correction and basis set variation for the B3LYP-D3 functional. Either theory underestimates the energy difference or the barrier for interconversion between methyl and phenyl docking is so low that it can still be surmounted when the complex is collisionally cooled to low temperature. We invite different theoretical studies with carefully balanced hydrogen bonding and dispersion interactions, such as DFT-SAPT combinations [43] or coupled cluster-based methods [44] to check whether they find a preference for methyl side docking in this parent complex. Given an estimated interconversion barrier from the phenyl to the methyl side of slightly more than 5 $\mathrm{kJ}/{\mathrm{mol}}^{-1}$ (B3LYP-D3), this preference should be on the order of 1 $\mathrm{kJ}/{\mathrm{mol}}^{-1}$ or even more to explain the experimental dominance of methyl docking in the jet expansion. For *ortho*-chloro- and *ortho*-bromo-acetophenone, evidence for stacking phenyl docking is overwhelming in both theory and experiment, and thus no surprise. For *ortho*-fluoro-acetophenone, there is evidence for a coexistence of all three docking variants. For *para*-halogenated acetophenones, the low IR visibility of the stacking isomers makes it difficult to derive rigorous abundance conclusions from the experimental spectra because it is even conceivable that the global minimum structure is not detected due to its weak IR absorption cross section, as emphasized in Figure 6. It would be instructive to record rotational spectra for such species [16], which allow for a more rigorous structure determination than the vibrational spectra and may be able to explore the energetical bistability of some of the systems presented in this work. Then, the same theoretical methods which succeed in describing the basic phenol-acetophenone balance should be tested with respect to their ability to describe the experimentally observed influence of the halogen atoms on the acetophenone. Methods which succeed for all systems are more likely to feature the correct balance between hydrogen, halogen, and generic dispersion interactions.

Some of these systems may be viewed as toy models for metamorphic proteins [45], which coexist in different folding states and can be tuned by small changes in the molecular sequence. Unlike the proteins, where bistability is energy- and entropy-driven, the intermolecular acetophenone–phenol balances are subtly dispersion-energy-controlled and serve as challenging benchmarks for the accuracy of quantum chemical methods.

## Figures and Tables

**Figure 1 molecules-26-04883-f001:**
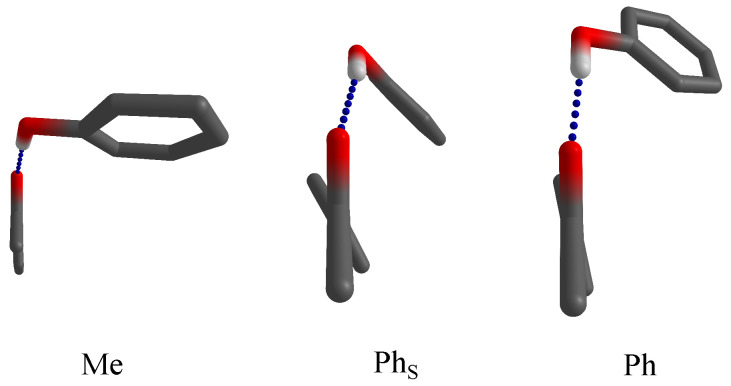
Docking variants for phenol on acetophenones (with the methyl group pointing toward the reader and the C=O group upwards). Methyl-side docking (Me) optimizes the hydrogen bond (blue dashes), whereas phenyl-side docking can either optimize the hydrogen bond (Ph) or phenyl stacking (${\mathrm{Ph}}_{S}$). The latter is sometimes assisted by a twist of the phenyl group in acetophenone.

**Figure 2 molecules-26-04883-f002:**
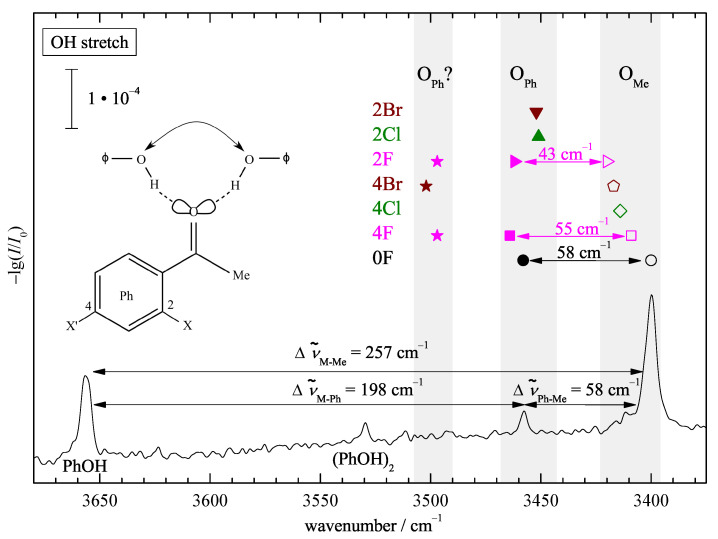
OH stretching FTIR spectrum of a coexpansion of phenol (PhOH) with acetophenone (0F) showing the phenolic OH stretching signals. The index Ph or Me indicates the docking side. The corresponding positions for halogenated acetophenones are summarized as symbols in grey corridors of 20 to 30 ${\mathrm{cm}}^{-1}$ for each side. Both docking sides are observed (double arrows) only for 0F, 2F, and 4F. See text for further explanations.

**Figure 3 molecules-26-04883-f003:**
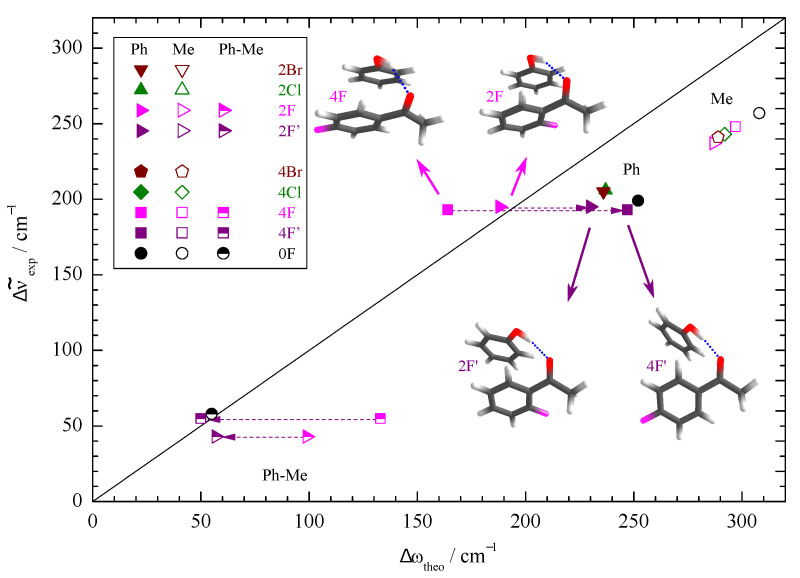
Correlation of the experimentally determined (anharmonic) OH complexation downshifts (Me, Ph) and splittings (Ph-Me) with their harmonic estimates predicted at B3LYP-D3/def2-TZVP level. The harmonic DFT shift overestimation, splitting prediction, and substitution trends are only uniform (symbol clustering) if the second-most stable Ph docking structures (2F’, 4F’) are assigned instead of 2F, 4F (dashed arrows).

**Figure 4 molecules-26-04883-f004:**
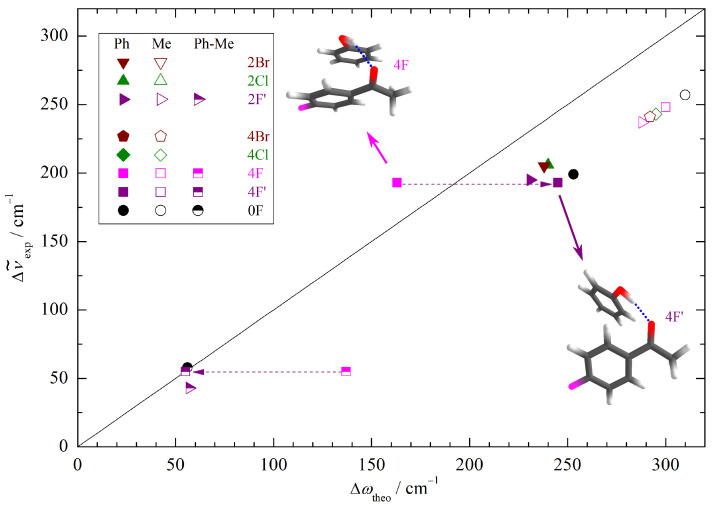
Correlation of the experimentally determined (anharmonic) OH complexation downshifts (Me, Ph) and splittings (Ph-Me) with the harmonic downshifts predicted at B3LYP-D3/ma-def2-TZVP level. The harmonic DFT shift overestimation, splitting prediction, and substitution trends are only uniform (symbol clustering) if the second-most stable Ph docking structure (4F’) is used for 4F (dashed arrows).

**Figure 5 molecules-26-04883-f005:**
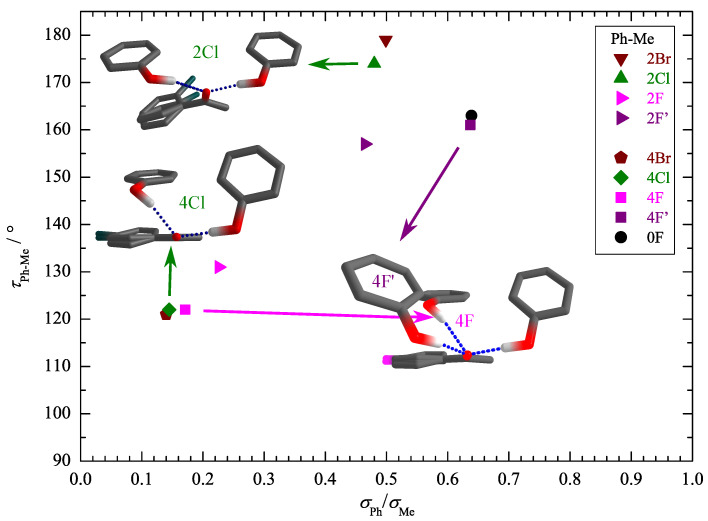
Correlation of the HO=CC${}_{\mathrm{Ph}}$ torsion angle difference for the two docking variants $\tau_{\mathrm{Ph}–\mathrm{Me}}$ (see Appendix A in the Supporting Information for the individual angles) with the theoretical intensity ratio $\frac{\sigma_{\mathrm{Ph}}}{\sigma_{\mathrm{Me}}}$. Strong out-of-plane placement in phenyl docking correlates with low IR visibility, whereas the methyl docking site behaves more uniformly.

**Figure 6 molecules-26-04883-f006:**
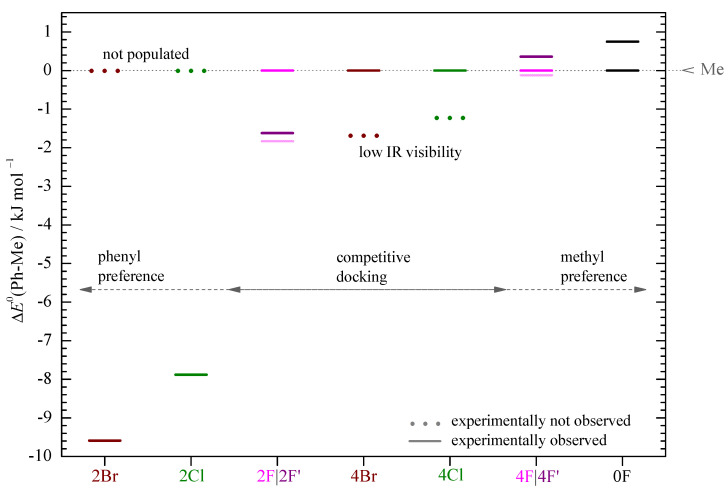
ZPE corrected energy differences $\Delta E^{0}$ (Ph-Me) in $\mathrm{kJ}/{\mathrm{mol}}^{-1}$ (B3LYP-D3(BJ,abc)/def2-TZVP) for each system referenced to the corresponding methyl sided coordination (Me) (see Appendix A in the Supporting Information) at $\Delta E^{0}=0\;\mathrm{kJ}/{\mathrm{mol}}^{-1}$. In 2Br and 2Cl, only the Ph sided structure and for the other systems at least the Me sided structure is experimentally observed (continuous line). The situation is unclear (light continuous line) only for 2F and 4F.

**Table 1 molecules-26-04883-t001:** Estimated fractions of methyl docking (Me), phenyl docking (Ph), and phenyl docking with dominant stacking (Ph${}_{S}$, low IR visibility if stacking competes with the hydrogen bond) for the phenol complexes of halogenated acetophenones in % of the total composition. Numbers in parentheses are generous estimates without unambiguous absorption feature, based on the signal size which the noise level may still hide.

Ketone	Me	Ph	Ph${}_{S}$	Phenol Preference
0F	81–89	19–11		Me
2F	29–52	<59	(<50)	Me, Ph
2Cl	(<11)		>89	Ph${}_{S}$
2Br	(<12)		>88	Ph${}_{S}$
4F	40–81	<30	(<52)	Me, Ph${}_{S}$
4Cl	>38		(<62)	Me, Ph${}_{S}$
4Br	>28		(<72)	Me, Ph${}_{S}$

## Data Availability

Most of the relevant data is contained within the article or Supporting Information. In addition, selected spectra and complex structures are provided in digital format (.dat, .xyz) under https://doi.org/10.25625/O01VGG (last access date: 8 August 2021).

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
