# Peer review of "Halogens in Acetophenones Direct the Hydrogen Bond Docking Preference of Phenol via Stacking Interactions"

_molecules, 2021, doi:10.3390/molecules26164883_

Round 1

Reviewer 1 Report

This paper by Zimmermann, Lange, and Suhm is an extremely fundamental study that describes the analysis of the hydrogen bonding/stacking preferences of phenol-acetophenones complexes by linear infrared spectroscopy and DFT calculations. The article is well written, and the discussion and conclusions were properly done based on the results presented. Even though this study does not have a huge potential to catch the eyes of the broad audience of “Molecules”, this piece of work contributes to the gigantic library of investigations on interactions involved in molecules probed by spectroscopy.

Author Response

see pdf

Reviewer 2 Report

The article entitled Halogens in acetophenones direct the hydrogen bond docking preference of phenol via stacking interactions written by C. Zimmermann et al is well written but it is quite dificult to follow. In order to improve this deficience I would like to suggest the authors the following points:
 - They should clearly present what the objectives of the article are.
 - I think a scheme with the two parent compounds that they are working with and where the substitutions and the different stacking ways occur would help readers.
-The same thing that I say about the objectives can be said about the conclusions of the work, I do not see that the study carried out allows us to conclude anything in particular. They should endeavor to clearly show the conclusions draw.

Author Response

see pdf

Reviewer 3 Report

This  paper  contains valuable  experimental  results   which could be useful in benchmarking  theoretical procedures.  However   authors   focused here on  the only one theoretical  method  based on  conventional  DFT  B3LYP-D3 approach,  without considering other alternatives.  Perhaps much more  appropiate teoretical  procedure to be  tested in this particular case  would  be    dispersionless DFT  augmented  by  much  more rigorous  dispersion  functions derived  from  state -of-the-art  Symmetry Perturbation Theory (SAPT)  (Podeszwa et al. J.Phys.Chem.Lett., 2010, 1, 550  https://dx.doi.org/10.1021/jz9002444   recently  extended  to  halogen  bonds (Jedwabny et al. J.Phys.Chem. A, 2020, 125,1787 https://dx.doi.org/10.1021/acs.jpca.0c11347?ref=pdf

Author Response

see pdf

Round 2

Reviewer 2 Report

Now, after the improvemnts made by the authors the article reaches the standards to be published.